# Dietary Factors in Sulfur Metabolism and Pathogenesis of Ulcerative Colitis

**DOI:** 10.3390/nu11040931

**Published:** 2019-04-25

**Authors:** Levi M. Teigen, Zhuo Geng, Michael J. Sadowsky, Byron P. Vaughn, Matthew J. Hamilton, Alexander Khoruts

**Affiliations:** 1Division of Gastroenterology, Hepatology and Nutrition, Department of Medicine, University of Minnesota, Minneapolis, MN 55455, USA; teige027@umn.edu (L.M.T.); geng0053@umn.edu (Z.G.); bvaughn@umn.edu (B.P.V.); 2BioTechnology Insititute, Department of Soil, Water & Climate, and Department of Plant & Microbial Biology, University of Minnesota, St. Paul, MN 55108, USA; sadowsky@umn.edu (M.J.S.); hami0192@umn.edu (M.J.H.)

**Keywords:** colon, high-sulfur foods, inflammation, metagenomics, microbiota, sulfur reducing

## Abstract

The biogeography of inflammation in ulcerative colitis (UC) suggests a proximal to distal concentration gradient of a toxin. Hydrogen sulfide (H_2_S) has long been considered one such toxin candidate, and dietary sulfur along with the abundance of sulfate reducing bacteria (SRB) were considered the primary determinants of H_2_S production and clinical course of UC. The metabolic milieu in the lumen of the colon, however, is the result of a multitude of factors beyond dietary sulfur intake and SRB abundance. Here we present an updated formulation of the H_2_S toxin hypothesis for UC pathogenesis, which strives to incorporate the interdependency of diet composition and the metabolic activity of the *entire* colon microbial community. Specifically, we suggest that the increasing severity of inflammation along the proximal-to-distal axis in UC is due to the dilution of beneficial factors, concentration of toxic factors, and changing detoxification capacity of the host, all of which are intimately linked to the nutrient flow from the diet.

## 1. Introduction

Inflammatory bowel disease (IBD), which includes both Crohn’s disease and ulcerative colitis (UC), is estimated to affect ~3 million individuals in the U.S. alone and continues to increase in incidence and prevalence worldwide [1]. Epidemiological evidence implicates industrialization, an increasingly western lifestyle, and associated changes in the microbiome with the development of IBD [2,3,4]. Diet is a major determinant of intestinal microbiota composition and activity [4,5]; therefore, it is a focus of intense interest for the mechanistic understanding of IBD pathogenesis. 

The risk of IBD development increases in immigrant children from developing countries [6] and substantial changes in the intestinal microbiome, including loss of diversity and displacement of *Prevotella* by *Bacteroides* strains, which can be observed following immigration from developing countries to the United States [7]. These findings lend strong support to the notion that changes in the gut microbiota, driven in part by diet, contribute to the pathogenesis of UC. Although attempts to identify consistent, specific ‘trigger’ foods that predict the development of UC have been unsuccessful, there appears to be an overall trend for an association between an animal-based diet and UC development and activity, while a plant-based diet may be protective against UC [8]. Conceptually, the epidemiological data that demonstrates an increasing prevalence of IBD with westernization lend support to this notion as westernization generally results in a transition from a plant-based to an animal-based diet [9]. 

The unique distribution of gut inflammation seen with UC, typically greatest in the rectum and extending continuously towards the proximal colon, supports the notion that a directionally concentrated toxic substance(s) may be involved in its pathogenesis. While the nature of this compound remains unknown, several lines of evidence, both observational and mechanistic, implicate hydrogen sulfide (H_2_S) as a possible candidate. The H_2_S toxin hypothesis, however, is complicated by the fact that H_2_S at low levels can be anti-inflammatory [10]. Consequently, the H_2_S toxin hypothesis may be better viewed in a concentration dependent manner where each individual host likely has a varied tolerance to a specific concentration of H_2_S above which intestinal damage occurs [11]. 

To date, attempts to examine the role of diet in the H_2_S toxin hypothesis have focused primarily on the intake of sulfur-containing foods, without regard for the overall diet composition [12,13,14,15]. It is important to consider, however, that outside of a laboratory setting diets are far more complex than can be captured by the measurement of a single or a few nutrients. Therefore, rather than focus on the specific intake of a single dietary component (e.g., sulfur) in the pathogenesis of UC, an emphasis should be placed on overall diet patterns (e.g., animal- vs plant-based diet). This approach has been the focus of a number of recent reviews [16,17], and preliminary data suggest that these types of compositional dietary changes may be beneficial in IBD [18].

The high protein content of animal-based diets provides a greater amount of sulfur that becomes available to distal gut microbiota [19]. Additionally, the higher fat content of a typical animal-based diet may lead to an increase in taurine conjugation to bile acids and a corresponding increase in the quantity of sulfur in the colonic lumen through an endogenous source [20]. Importantly, an animal-based diet pattern will also tend to be low in fiber, which may drive a metabolic shift of microbiota away from carbohydrate and towards protein fermentation and increased mucin degradation (Figure 1) [21,22].

A complimentary hypothesis to explain the directional biogeography of UC may be the dilution of beneficial microbial metabolites along the luminal flow in the intestine. One such set of products is the short-chain fatty acids (SCFA), which are generated as a result of complex polysaccharide fermentation and play essential roles in colon physiology and mucosal homeostasis. Thus, butyrate (a SCFA), in addition to being a preferred energy source for the colonocytes [23,24], also promotes the presence of regulatory T cells in the colon [25,26], strengthens the gut barrier function [27], decreases epithelial oxygenation [28,29], and inhibits excessive colonic stem cell proliferation [30]. Furthermore, the chemical milieu established by microbial metabolites impact the overall microbial community structure through feedback loops affecting the relative abundances of different microbial trophic groups. Lower pH, driven by higher concentrations of SCFA, favors the growth of methanogens over that of sulfate reducing bacteria (SRB) [31], whereas H_2_S may increase the oxygenation in colonocytes by inhibiting β-oxidation [32] and lead to the inhibition of obligate anaerobes that produce SCFA.

Therefore, the updated understanding of the H_2_S toxin hypothesis for UC pathogenesis presented here strives to account for the activity of the *entire* microbial community rather than individual microbial subgroups working in isolation (e.g., *Desulfovibrio*) and appreciates the interdependency of diet composition on both the structure and function of the microbial community. In many respects, this is a synecological approach to study of the relationship between gut microbiota and the development of UC.

### 1.1. Toxic Effects of Hydrogen Sulfide

Although H_2_S is generated by the host and is increasingly recognized to have a multitude of important beneficial physiologic functions [33], it becomes a potent toxin once its concentration exceeds the detoxifying capacity in the tissue. Specifically, higher amounts of H_2_S generated in the intestine have the potential to disrupt the gut barrier function, which may be an early and critical initiating event in triggering the onset of UC and the perpetuation of its activity [34]. Traditionally, the H_2_S toxin hypothesis has focused on the potential injurious effects of sulfide gas on the cellular metabolism of colonocytes, mainly the inhibition of cytochrome *c* oxidase activity in mitochondria, which induces oxidative stress in a fashion similar to cyanide [35]. Roediger and colleagues demonstrated that H_2_S inhibits β-oxidation of butyrate by colonocytes, their preferred energy source [36]. Notably, UC mucosa has lower rates of butyrate uptake and oxidation relative to healthy controls [37]. In summary, oxidative stress and energy starvation caused by excessive H_2_S concentrations may lead to colonocyte death, penetration of the epithelial barrier by the intestinal microbes and their direct interaction with the mucosal immune system. Resulting inflammation leads to further disruption of the gut barrier, decreased butyrate oxidation [37], and decreased mucosal sulfide detoxification and the subsequent perpetuation of inflammation [38]. 

In recent years, considerable attention has also been focused on mucus integrity in UC. The mucus layer of the colon consists of an outer (loose) and an inner (dense) layer, the latter being largely impenetrable to the resident intestinal microbiota [39,40]. The inner mucus layer in UC patients and in some animal models of IBD is more penetrable to bacteria relative to healthy controls [41]. Mucin glycoproteins form the major building blocks of mucus. The dominant mucin secreted by the goblet cells in the colon is MUC2, which contains cysteine-rich domains that mediate its multimerization through disulfide bonds. H_2_S can directly break the sulfide bonds in the mucus and disrupt the MUC2 network leading to a loss of mucus viscosity and greater permeability [42]. 

### 1.2. Pre-Clinical Models

Some of the strongest support for the H_2_S toxin hypothesis in the literature comes from pre-clinical models. The most commonly used animal model of IBD uses dextran sodium sulfate (DSS) to induce damage to the epithelium [43]. This allows for bacterial translocation from the lumen of the intestine resulting in inflammation [43]. This model is renowned for its simplicity and consistency in producing epithelial damage similar to that seen in IBD and also lends support to a role of sulfur in the pathogenesis of UC in humans. 

In IL-10 knockout mice, a common murine model of IBD, a diet high in saturated fat increases the presence of taurine-conjugated bile acids, leading to an expansion of the sulphite-reducing pathobiont *Bilophila wadsworthia* and a greater severity of inflammation [20]. High fat diets also promote intestinal inflammation in rats as well as adenoma formation in the presence of a mutagen [44]. A high protein diet in a similar model increases SRB abundance and sulfide production and decreases the abundance of bacterial taxa associated with SCFA production and the amount of butyrate in stool [45]. Similarly, a high protein diet has been associated with post-weaning diarrhea in piglets [46]. Taken together, a high protein and saturated fat diet (characteristic of a western, animal-based diet) seems to result in an increased capacity for H_2_S production, decreased capacity for butyrate production, and a potential to cause intestinal inflammation. 

### 1.3. Clinical Observations

Mesalamine, a first-line agent in treatment of mild-to-moderate ulcerative colitis, inhibits sulfide production in a dose-dependent manner when added to a fecal slurry *in vitro* [47]. Furthermore, UC patients taking mesalamine have reduced fecal sulfide relative to patients who do not [47]. It is noteworthy that mesalamine and butyrate are both peroxisome proliferator activated receptor gamma (PPAR-γ) agonists [48,49]. Stimulation of PPAR-γ promotes β oxidation of fatty acids and epithelial hypoxia in the colon, which favors obligate anaerobes and lowers the abundance of Proteobacteria [50]. Interestingly, some antibiotics (e.g., aminoglycosides) which target Proteobacteria (including SRB) have short-term benefits in UC patients [13,51]. 

Few studies have attempted to link dietary interventions targeting the updated H_2_S toxin hypothesis (i.e., transition from a western-style diet) with UC. The available handful of reports is limited to small case series or individual case studies (Table 1) [52,53,54,55,56]. These low dietary sulfur interventions generally emphasize transitions to a plant-based, semi-vegetarian diet. While all these experiences describe positive outcomes, conclusions are limited given the small patient numbers. The earliest case series conducted by Roediger et al. included only four patients and relied on symptom and histological criteria [52]. These patients sustained improvement in the activity of UC over 18 months of follow-up, while being maintained on the low sulfur diet. Chiba et al. reported a decreased relapse rate of UC relative to historical expectations among 60 patients instructed to follow a plant-based diet and followed for up to five years [56].

## 2. Sources of Sulfur in the Colon

### 2.1. Dietary Intake

While sulfur is ubiquitous in the food supply [57], and the fifth most abundant element on earth, its dietary linkage is not well characterized. For example, the USDA Food Composition Database does not include sulfur as a searchable nutrient. The sulfur content of food can be estimated using the sulfur-containing amino acids (methionine and cysteine) as surrogates, but this fails to account for sulfur-containing food modifiers or additives, such as sulfiting agents (e.g., potassium bisulfate, sodium bisulfate), sulfuric acid, or carrageenan [58]. Therefore, the most robust method of estimating dietary sulfur intake, including an estimate for inorganic sulfur, is through a 24–48 hour urine collection [59,60]. Because the primary dietary sources of sulfur are the sulfur-containing amino acids, in practice, quantification of sulfur-containing amino acid content has been used to create low- and high-sulfur diets [52,61]. 

Dietary input and small intestinal absorption are considered the main determinants of sulfur delivery to the colon [19]. The small intestine has an efficient but saturable dietary sulfate absorptive capacity of ~5–7 mmol/day [62]. Once dietary sulfate intake exceeds ~5–7 mmol/day the amount reaching the colon increases linearly with intake. Therefore, the amount of dietary sulfur that reaches the colon, particularly with increasing levels, can be generally assumed to parallel dietary intake. A number of additional factors, however, such as food preparation (e.g., cooked versus uncooked, cooking temperature and method, ground versus whole), meal consumption habits (e.g., chewing), and transit time have been shown to influence small intestinal absorption of sulfur-containing molecules and the total amount of sulfur reaching the colon [63,64,65,66,67].

### 2.2. Endogenous Sources

The main endogenous sources of sulfur are taurine-conjugated bile acids and mucin. Taurine is a sulfur-containing amino acid that is synthesized in the liver from methionine and cysteine. Diet can influence the rate of taurine conjugation and subsequent presence of taurine conjugated bile acids in the colon [68]. Although the majority of bile acids are re-absorbed in the distal small intestine into the enterohepatic circulation, a small percentage will spill into the colon [69]. Deconjugation of bile acids is an early step in secondary bile metabolism and is mediated by gut microbiota. 

Intestinal mucins are glycoproteins that form a mucus barrier along the epithelial surface of the gastrointestinal tract. The primary intestinal mucin MUC2 contains cysteine amino acids as core components of its structure [39,70]. Glycosylation of mucin proteins makes up a substantial portion of their size and provides protection of the peptide backbone and the gel-forming capacity [70]. The glycosylation pattern of mucin varies along the intestinal tract with increasing sulfation moving distally through the small intestine and into the colon [71,72]. The colonic mucus barrier is predominately composed of sulfomucins with a trend toward slightly diminished sulfation moving distally (100% sulfation in the right colon and 86% in the rectum) [73]. The normal sulfation pattern is altered in UC and is associated with the concentration of SRB [73]. 

### 2.3. H_2_S Measurement

The ideal experimental system to test the role of H_2_S in UC pathogenesis would be able to measure its intraluminal concentration at different segments in the intestine. Unfortunately, technology to do that does not yet exist; therefore, studies that measure H_2_S have focused on ex-vivo determination of its production from fecal samples. Fecal concentrations of hydrogen sulfide, however, are notoriously difficult due to the volatility of the gas, low intestinal concentrations, and instrumentation issues. Historically, the majority of work assessing stool sulfide concentrations in UC relied on colorimetric assays [62,74,75]. This method requires ‘trapping’ H_2_S in fresh fecal samples for measurement and is associated with a number of inherent limitations [76,77]. Ion-exchange chromatography has also been used to measure H_2_S in feces [78]. 

H_2_S gas measurements, done using gas chromatography (GC) with sulfur chemiluminescence detection, are considered the reference standard for H_2_S measurement [79]. These measures, however, cannot be obtained in real time and require *in vitro* incubation of the fecal sample. Only one study of UC has used GC as the measurement method and found that H_2_S production was elevated substantially in UC compared to controls over a 24 hour period [80]. Although this method does not capture the H_2_S content of fresh fecal samples, *in vitro* incubation may reflect *in vivo* H_2_S production capacity of the microbiota, providing a *functional* measure of the microbiota metabolic capacity. Furthermore, this method allows for interrogation of the capacity of various dietary components to affect H_2_S production (e.g., more H_2_S production with cysteine or mucin vs sulfate) [80,81].

## 3. Colonic Microbiota and H_2_S Production

### 3.1. Sulfate-Reducing Bacteria 

Microbial production of H_2_S is generally thought to be carried out by a limited number of bacteria and archael species. Sulfate-reducing bacteria (SRB) and the dissimilatory sulfur cycle, a form of anaerobic respiration that uses sulfate as the terminal electron acceptor, have been the primary focus of the H_2_S toxin hypothesis in UC [14]. This pathway generates H_2_S as an end-product of sulfate reduction. The pathway consists of three enzymatic steps involving ATP sulfurylase, adenosine 5’-phosphosulfate reductase, and sulfite reductase [82]. The final step in the pathway, sulfite reduction, is considered the rate limiting step. The genus *Desulfovibrio* is regarded as the most abundant SRB in humans [83]. Despite the focus on *Desulfovibrio* spp. as the primary producer of H_2_S in the human gut, the number of microbial groups known to be capable of dissimilatory sulfate reduction continues to expand [82,84]. Many of these microbes can also reduce sulfite, dithionite, thiosulfate, elemental sulfur, and several thionates. The three sulfite reductase genes which catalyze the production of hydrogen sulfide are dissimilatory sulfite reductase (dsr), anaerobic sulfite reductase (asr), and cytochrome *c* sulfite reductase (*mccA*) [82]. 

In humans, the measurement of key enzymes involved in the sulfate reduction pathway is a more specific method for measurement of SRB than measurement of bacterial genera such as *Desulfovibrio* [85]. This approach identified a unique capacity of SRB genera in UC to generate H_2_S and induce epithelial apoptosis, compared with healthy controls [74,86]. Recent multiomics data in colon cancer lends additional support to the importance of accounting for the functional capacity of microbes (e.g., via measurement of key enzymes) in addition to their community composition [87].

Despite their importance and usefulness, there is a dearth of literature that relies on the quantification of sulfite reductase genes to characterize SRBs in large UC cohorts. Use of this technique in healthy subjects demonstrated that an animal-based diet increases sulfite reductase gene expression [5]. Recent work by Jia et al. used a semi-nested PCR method to detect *dsrB* DNA, but failed to find a difference in abundance between healthy and UC cohorts (*n* = 18 and *n* = 14, respectively) [88]. The agar shake dilution method, which measures SRB growth over a specified incubation period, has often been employed to measure SRB in individuals with UC. Studies done using this methodology found elevated SRB abundance in individuals with UC compared to those without [47,74,89]. 

### 3.2. Cysteine Degraders

Although sulfate intake and quantity of SRB has been the primary focus of the H_2_S toxin hypothesis, there is a growing appreciation for the contribution of protein intake to H_2_S production. The efficiency of protein degradation is greatest in the distal colon and at neutral to alkaline pH, which suggests a possible relationship with the pathogenesis of UC [90]. A cross-over diet study conducted by Magee and colleagues demonstrated that dietary protein intake positively correlates with fecal sulfide concentrations in healthy individuals [78]. 

A recent *in vitro* study that profiled gas production from incubated fecal samples implicated cysteine specifically as a primary driver of H_2_S production—conversely, the effect of sulfate was small [81]. Cysteine degradation to H_2_S is catalyzed by enzymes with cysteine desulfhydrase activity. Although the enzymes with cysteine desulfhydrase activity are well characterized in humans [91], identifying enzymes with this activity in microorganisms has proven far more challenging. *Escherichia coli* possess several enzymes with cysteine desulfhydrase activity, including *O*-acetylserine sulfhydrylase-A, *O*-acetylserine sulfhydrylase-B, MalY, tryptophanase and cystathionine β-lyase [92]. Because cysteine likely makes a substantial contribution to H_2_S production in the colonic lumen, there is a critical need to detail cysteine desulfhydrase activity within the colonic microbiome.

Prevention of excessive cysteine degradation to H_2_S may underlie the possible protective role of appendectomy for appendicitis in UC [93]. Although the mechanism underlying the purported beneficial effect of appendectomy in UC has yet to be elucidated, one possibility is related to the potential role of the appendix as a reservoir for gut microbes [93]. Specifically, the appendix contains *Fusobacterium* spp. in its healthy state [94] and appendicitis is associated with an abundance of this bacteria [95]. *Fusobacterium* spp. contain a number of proteins necessary to metabolize L-cysteine to H_2_S [96,97]. Therefore, removal of the appendix, and subsequently a source of H_2_S production, may contribute to its possible therapeutic role in the prevention of UC.

### 3.3. Interplay Between Functional Microbial Groups 

A number of physicochemical factors shape the composition and functional output of microbial communities, which need to be integrated in order to fully account for the inter-individual differences. One such factor is the availability of hydrogen (H_2_), which is essential as a substrate for anaerobic respiration in the colonic lumen. The groups of bacteria that rely on and compete for H_2_ for anaerobic respiration are acetogens, methanogens, and SRB Figure 2) [98]. H_2_ production results from microbial fermentation of carbohydrates in the lumen of the intestine. Therefore, the balance between H_2_ production and consumption is often referred to as the ‘hydrogen economy’ [99]. An imbalance of H_2_ consumption and production may result in a metabolic environment that consumes NADH for H_2_ production at the expense of butyrate production [100]. In the context of the H_2_S toxin hypothesis, it is important to consider possible imbalances in H_2_ consumption (e.g., altered SRB concentration) or production (e.g., low dietary intake of fermentable carbohydrate) that could create a metabolic environment leading to UC—namely, excessive H_2_S production [80], mucin degradation [22,41], and diminished butyrate production [5]. 

Other physical (e.g., intestinal transit time) and chemical factors (e.g., pH and oxygen tension), also contribute to the microbial community structure in the colon by favoring some groups of microbes over others. Accelerated colon motility is likely to benefit fast-growing microorganisms and disfavor slow growing microorganisms, and constipation is associated with higher carriage of methanogenic archaea and increased methane production [101]. Fermentation of digestible carbohydrate results in the production of SCFA and a drop in luminal pH in the colon relative to the small intestine [102,103,104]. Increased dietary fiber also results in faster intestinal transit [105]. Indeed, the colon pH is lowest proximally and increases distally, which is likely driven by the decreasing availability of digestible carbohydrate [102]. Mildly acidic pH, characteristic of the proximal colon, is inhibitory to the growth of SRB, but favors the growth of methanogens and some butyrate producers [30,106,107]. In contrast, sulfate reduction and H_2_S production are optimal at an alkaline pH [108].

### 3.4. H_2_S Clearance

Clearance of hydrogen sulfide in the gut can be viewed as the final component of the H_2_S toxin hypothesis. Due to its potential deleterious effects, H_2_S is highly regulated in the cell. As H_2_S is freely permeable across membranes, luminal concentration in the colon would be expected to correlate with intracellular concentrations [10]. Recent work in animal models demonstrates the relationship between luminal H_2_S and intracellular regulation [109]. Although a small amount of H_2_S present in the lumen of the colon is able to be cleared through flatus [110], the primary pathway for H_2_S clearance is via intracellular sulfide oxidation. Interestingly, this pathway may be facilitated by cyanide, which has been proposed by Levitt and colleagues to explain the well-documented beneficial role of smoking in UC [111].

Colonic mucosa possesses a very efficient system for H_2_S detoxification, and defects in these pathways are hypothesized to contribute to pathogenesis of colitis [111,112]. In earlier work, thiol methyltransferase and rhodanese were thought to have a major role in the detoxification of intestinal H_2_S [113]. However, the investigators found conflicting results when using these enzymes to estimate the H_2_S detoxification capacity in UC [38,114]. Around the time this work was being conducted, however, a more detailed and comprehensive understanding of H_2_S detoxification was being developed that implicates sulfide quinone oxidoreductase (SQR) as the rate-limiting enzyme in sulfur oxidation [115,116]. 

The SQR enzyme is located within the inner mitochondrial membrane [117]. The reaction of H_2_S with SQR results in oxidation of H_2_S to a small-molecule persulfide and reduction of coenzyme Q (CoQ), which links H_2_S oxidation with complex III of the electron transport chain [117]. The sulfur transfer from H_2_S to a small molecule acceptor occurs via a cysteine intermediate within the SQR enzyme. Therefore, the availability of a small molecule acceptor is believed to be the rate-limiting step in the SQR reaction [118]. The antioxidant glutathione (GSH) is also believed to be the primary acceptor in a physiological setting, but sulfite has also been proposed as an alternative acceptor [118]. If GSH is utilized as the acceptor, the result is formation of glutathione persulfide (GSSH), which is then utilized by persulfide dioxygenase (ETHE1) or rhodanese to form sulfite or thiosulfate, respectively [117,119]. If sulfite is used as the acceptor, thiosulfate is produced, and rhodanese instead converts thiosulfate to GSSH, which is then converted to sulfite by ETHE1 [117,119]. In a steady state, GSH has been estimated to be favored ~5:1 over sulfite [118], but elevated intracellular sulfite concentrations or depletion of GSH have been suggested to inhibit SQR [116,118,119]. Furthermore, although H_2_S oxidation appears to be a robust pathway for intracellular control of H_2_S levels, an excess of H_2_S may paradoxically result in feedback inhibition of SQR from the electron transport chain as a result of its deleterious effect on cytochrome *c* oxidase activity [119].

The H_2_S detoxification capacity and expression of enzymes involved in the H_2_S detoxification pathway (SQR, ETHE1, and thiosulfate sulfurtransferase) have been shown to follow a general trend of highest in the proximal colon to lowest in the rectum [120]. This expression pattern lends further support to the H_2_S toxin hypothesis as the progression of UC progresses from the lowest to the highest H_2_S detoxification capacity and enzyme expression (Figure 3). In addition, GSH synthesis has been demonstrated to be impaired in IBD [121]. 

## 4. Dietary Factors Influencing Colonic H_2_S Production

### 4.1. Protein

Dietary protein has been shown to be a robust driver of *in vitro* and *in vivo* H_2_S production [78,81]. Furthermore, the efficiency of protein degradation is inverse to that of carbohydrate fermentation (and SCFA production) and parallels that of UC pathogenesis (highest in the distal colon and decreasing proximally) [90]. Dietary protein is efficiently absorbed in the small intestine, but the amount reaching the colon is thought to be positively correlated with the amount consumed [90]. 

In both human and animal studies, a high protein diet results in fecal microbiota changes that increase H_2_S production and decrease SCFA production [5,45]. Increases in SRB and *Bacteroidetes* sp. observed in UC also occur in a high protein diet [5,45,122] and observed decreases in specific butyrate producing species within a high protein diet (*F. Prausnitzii, Akkermansia, and Ruminococcus*) mirror those seen in UC [45,122,123,124]. 

Sulfur-containing amino acids are the primary source of dietary sulfur. Because of limited information on the sulfur content of food, outside of a metabolic unit, a ‘high’ or ‘low’ sulfur diet is defined according to cysteine and methionine concentration [52]. Therefore, a ‘low sulfur’ diet is actually a ‘sulfur-containing amino acid-controlled diet’, rather than a true ‘sulfur-controlled diet’ [52]. In general, a sulfur-containing amino acid-controlled diet results in a shift from a more traditional western diet (high in animal protein and fat, and low in fiber) to more of a plant-based diet (high in fiber, low in animal protein and fat). Recently published diet composition data from a small sulfur-containing amino acid diet intervention study (*n* = 4) conducted in the U.S. suggests that a typical U.S. diet is already high in sulfur-containing amino acids; a transition from a baseline diet to a ‘high sulfur’ diet results in minor diet composition changes, while a transition from a baseline to a ‘low sulfur’ diet results in a substantial transition from an animal-based to a plant-based diet (resulting in increased fiber, and decreased protein and fat intake) [61].

As described above, H_2_S production capacity is often characterized by quantifying SRB, but cysteine desulfhydrase activity provides another route of H_2_S production for the colonic microbiota that is often left unassessed. In fact, *in vitro* modeling using healthy human feces suggests that the contribution of cysteine to H_2_S production is substantially more than that from sulfate [81]. 

### 4.2. Carbohydrate and Fiber

High carbohydrate availability in the colon, as noted above, promotes microbial groups able to utilize carbohydrate substrates, but also affects other aspects of microbial metabolism and especially impacts protein degradation. The addition of fermentable fiber to healthy human feces in an *in vitro* setting drastically reduces H_2_S production from any source (e.g., cysteine, sulfate) [81]. Lower pH associated with microbial carbohydrate fermentation also leads to inhibition of dissimilatory aromatic amino acid metabolism [125]. Interestingly, stool pH is lower in vegan and vegetarian individuals relative to omnivores, consistent with greater abundance of carbohydrates in the proximal colon, faster colon transit, and higher delivery of SCFA to the distal colon [126]. This finding underscores the importance of overall diet composition when considering the relative proportions of different end-products of microbial metabolism.

### 4.3. Dietary Fat

Although there is little data supporting a direct role of dietary fat in the H_2_S toxin hypothesis, animal fat was shown in a murine model to result in increased production of taurine-conjugated bile acids and a bloom of the sulfite-reducer *Bilophila wadsworthia* in the colonic microenvironment [127]. A high fat diet in mice is associated with lesser production of SCFA and greater production of H_2_S, even when the protein content of high fat chow is lower relative to regular chow [128]. It is obvious that a relative contribution of macronutrients to H_2_S production is difficult to isolate in a single comparison, given that an increase in one will correspondingly necessitate reduction in at least one other to maintain the same caloric intake. 

### 4.4. Endogenous Sulfur

The most significant source of endogenous sulfur (both as sulfate and cysteine) is intestinal mucin. The mucous layer of the colon is composed of two layers [40], the microbiota is abundant in the outer loose mucous layer, while the inner dense mucous layer—largely devoid of bacteria—maintains a barrier between the colonic microbiota and colonocytes [39,40]. The composition of the colon microbiota has been shown to influence mucin secretion, breakdown, structure, and the glycosylation pattern [22,42,129,130,131]. Conversely, mucin glycosylation patterns have been shown to influence the colonic microbiota [132]. Therefore, the H_2_S toxin hypothesis must consider these complex relationships between the mucin, microbiota, and diet. A characteristically high sulfur diet (high in animal protein and fat, and low in fiber) leads to nutrient deprivation for the microbiota and results in increased mucin desulfating sulfatase activity and mucin degradation [5,22,131,133]. This pattern is remarkably similar to that seen in UC [134,135,136]. The phenotypic changes resulting in increased mucin breakdown (e.g., low fiber) and H_2_S production (e.g., high protein/sulfur intake) [5,22,45,78] may produce H_2_S concentrations that overwhelm the detoxification capacity of colonocytes and contribute to H_2_S toxicity. 

### 4.5. Miscellaneous Dietary Factors 

Finally, it is also important to consider factors that may correlate with sulfur content in the diet but are not related to sulfur metabolism. An increased intake of heme, present in its highest concentrations in red meat, can have cytotoxic effects on colonocytes [42]. Consumption of processed foods expose individuals to additives such as phosphates, nitrates, and emulsifiers, which have been shown to influence the composition of microbiota, mucin layer thickness, and intestinal inflammation [137,138,139,140]. Even specific cooking methods may alter the potential of diet to impact the intestinal microbiome and microbiota-host interactions. Both enzymatic and non-enzymatic mechanisms of protein modification may also influence intestinal H_2_S production. Thus, glycation of dietary protein can to shift the colonic microbiota towards greater abundance of SRB and lesser production of butyrate [64]. Conversely, there are likely undiscovered or underappreciated components of a plant-based diet that may be protective in UC [141].

### 4.6. Inflammatory Factors

Onset of inflammation triggers a number of positive feedback loops that amplify the detrimental effects of H_2_S on the homeostasis of colonic mucosa. Increased synthesis of inducible nitric oxide synthetase (iNOS) results in increased release of nitrate, which contributes to dysbiosis by encouraging the expansion of facultative anaerobic Proteobacteria [28]. Nitric oxide also impairs H_2_S detoxification [109], thus lowering the threshold concentration where H_2_S becomes toxic to epithelial cells. Epithelial injury results in crypt hyperplasia and loss of epithelial hypoxia, which are detrimental to butyrate-producing obligate anaerobes. In animal models and human studies, SCFA (and specifically butyrate) production is inhibited in the presence of inflammation [122,123,124,142]. Inflammation is also correlated with a lower BcoAT gene content of fecal microbiota, consistent with lower butyrate production capacity [143]. Collectively, these changes hasten the collapse of the gut barrier to intestinal microbes and inhibit regulatory immune circuits, such as regulatory T cells, which further augment the inflammatory response. 

### 4.7. Interventions Targeting the Colon Microbiota 

The underlying mechanisms that make up the H_2_S toxin hypothesis are supported by changes produced with fermentable fiber supplementation in individuals with UC (increased butyrate production) [144] and findings from a recent fecal microbial transplantation (FMT) trial that demonstrated a correlation between sustained remission and increased butyrate production, while increased abundances of Bacteroidetes and Proteobacteria correlated with no response or relapse [122]. Notably, members of the Bacteroidetes phylum contain mucin desulfating sulfatase enzymes [122,145,146]. The FMT in UC findings have also been recently expanded upon by Wei et al. who demonstrated a role for fiber in maintaining the gut microbiota composition in individuals with UC following FMT [147].

The role of carbohydrate, and more specifically fermentable carbohydrate, intake within the H_2_S toxin hypothesis may be its overall impact on protein degradation by colonic bacteria. These findings are supported by a diet study in healthy volunteers [148], where administration of metronidazole effectively reduced SRB counts, but fecal concentration of H_2_S remained unchanged. In contrast, consumption of fermentable fiber (oligofructose) resulted in a decrease in fecal H_2_S, concomitant increase in SCFA concentrations, but failure to influence SRB counts. The impact of fiber on microbial H_2_S production is further supported by recent data demonstrating a decrease in SRB with a high brassica diet (sulfur-containing vegetables) [149]. Although brassica vegetables are technically considered to be high sulfate foods, they also have a high fiber content. These findings underscore the beneficial role for the latter within the H_2_S toxin hypothesis and support a clinical approach focusing on plant versus animal-based diets rather than high- and low-sulfur diets. 

It is important to consider that while short-term changes in diet can affect activities of different microbial groups in the intestine, they do not change the overall individual-specific taxonomic composition of microbiota [61,150]. Short-term low carbohydrate, high protein diets are among the most popular approaches used to achieve weight loss. Such diets do result in decreases in relative abundance of some butyrate producing bacteria, as well as altered fecal metabolite profiles, including lower content of SCFA and fiber-derived antioxidant phenolic acids [151]. It is possible that such dietary changes can even trigger the onset of UC, which may be reversed with increasing plant-derived dietary components [54,56]. However, long-term diets low in microbiota-accessible carbohydrates can lead to complete extinctions of microbial taxa [152]. Therefore, it is possible that most consistent benefits of dietary therapies will also require transplantation of relevant microbiota, optimized in capacity for the individual patients.

## 5. Conclusions

There is an intriguing body of literature supporting a relationship between dietary sulfur intake and the H_2_S toxin hypothesis in the development of UC. To date, the focus of the H_2_S toxin hypothesis in UC has primarily been focused on the abundance and activity of SRB (e.g., *Desulfovibrio*) and dietary sulfur intake. However, this provides only a partial window on the total activity of the microbiota that is relevant to UC pathogenesis. An updated approach to the H_2_S hypothesis has to incorporate interactions between different microbial groups as well as multiple components of the diet. A compelling case does exist that a typical western diet, which tends to be high in protein and low in fermentable fiber, may promote the accumulation of harmful products such as H_2_S and even increase their toxicity, and is also associated with a lower production of beneficial products such as butyrate. Patients are intensely interested in adjunctive dietary therapies for UC. Multiple technological developments that allow compositional and functional measurements of entire microbial communities should facilitate the development and testing of novel microbiota-targeting interventions that include the dietary management of IBD. 

## Figures and Tables

**Figure 1 nutrients-11-00931-f001:**
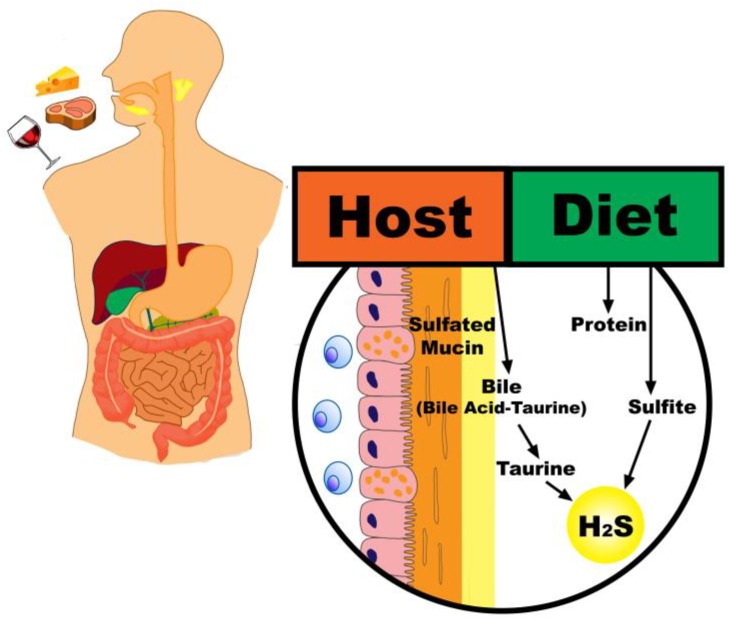
Contributions of an animal-based diet to hydrogen sulfide production. Legend: An animal-based diet results in a greater amount of dietary sulfur available to distal gut microbiota, both directly through sulfur-containing amino acids and indirectly through an increase in taurine conjugated bile acids and mucin degradation.

**Figure 2 nutrients-11-00931-f002:**
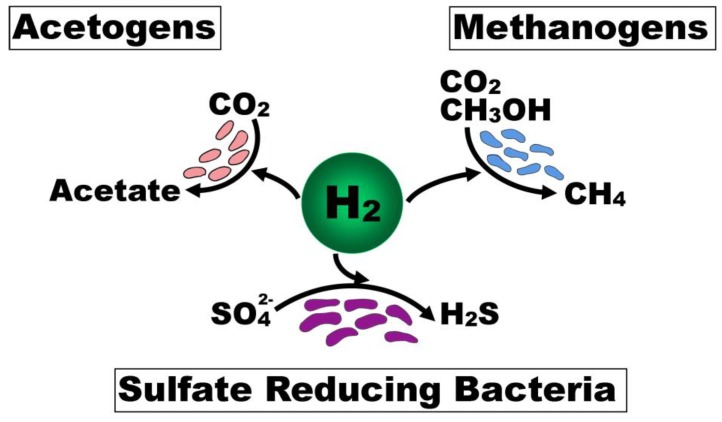
Bacterial competition for hydrogen for anaerobic respiration in the lumen of the intestine. Legend: Acetogens, methanogens, and sulfate reducing bacteria are the microbial groups that compete for H_2_ in anaerobic respiration in the lumen of the colon. The availability of hydrogen can shape the composition and functional output of microbial communities.

**Figure 3 nutrients-11-00931-f003:**
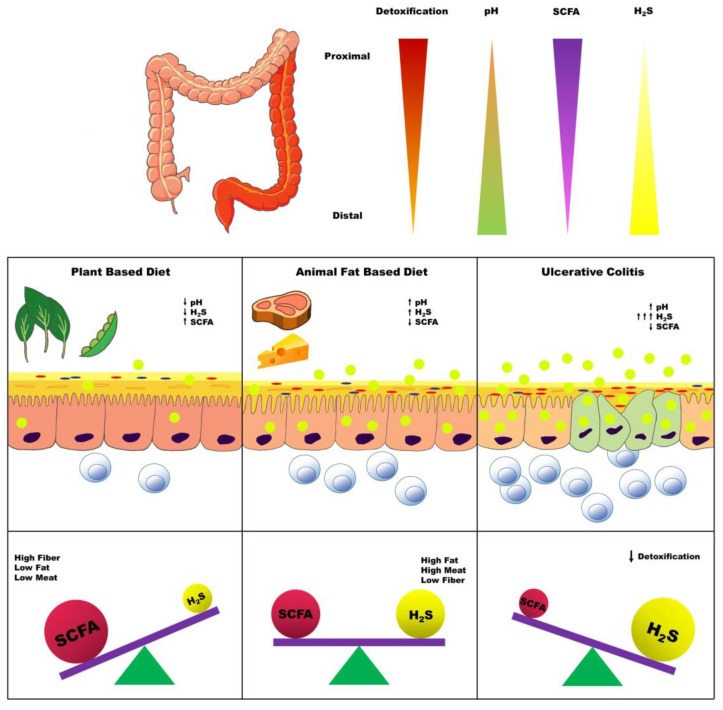
Overview of metabolic milieu and influence of diet in the pathogenesis of ulcerative colitis. Legend: The distribution of inflammation seen with ulcerative colitis is typically greatest in the rectum and extends continuously towards the proximal colon with varying severity. This pattern of inflammation parallels decreasing expression of the host H_2_S detoxifying enzymes, rising pH, and decreasing concentrations of short-chain fatty acids (SCFA). A plant-based diet promotes greater production of SCFA, but not H_2_S. SCFA production decreases in an animal-based diet, while H_2_S production increases; however, there is a sufficient capacity for H_2_S detoxification to prevent epithelial cell damage. Nonetheless, a decreased capacity for H_2_S detoxification results in inflammation, which in turn further exacerbates the gradients of SCFA and H_2_S concentrations along the proximal-to-distal colon axis.

**Table 1 nutrients-11-00931-t001:** Summary of reports detailing dietary interventions in the treatment of ulcerative colitis (UC) targeting the updated H_2_S hypothesis.

Author Year	Location	Study Design	Intervention	Outcomes
Roediger 1998 [52]	Australia	Prospective pilot, Single-arm, 12 months duration (*n* = 4)	Reduction of sulfur amino acid intake for 12 months following an acute attack of UC	Histological improvement at 12 months. Decreased number of daily bowel movements and stool more formed. Two patients noted worsening symptoms when off diet.
Kashyap et al. 2013 [53]	United States	Case Report 26-year-old male with notable history of migration to Canada at the age of 10 from Southeast Asia and transition to western diet	Eliminated dairy, refined sugar, pork, beef, and fried food. Emphasized vegetables, fruit, chicken, and fish.	Majority of UC symptoms alleviated at 12 months. Prospectively, symptoms worsened when off diet and improved when returned to diet.
Chiba et al. 2016 [54]	Japan	Case Report 38-year-old male who began an Atkins-type diet that emphasized meat and animal fat 2-years prior to UC diagnosis	Transitioned to plant-based, semi-vegetarian diet	Achieved remission without medication. Qualitative comment that symptoms returned with worsening diet compliance over time.
Chiba et al. 2018 [55]	Japan	Case Report 36-year-old female diagnosed with UC at 21 weeks gestation who experienced notable diet changes with onset of emesis	Transitioned to a plant-based, lacto-ovo-vegetarian diet	Remission achieved and maintained through pregnancy
Chiba et al. 2018 [56]	Japan	Prospective, single-arm (*n* = 60)	Individuals with mild UC or UC in remission were provided with plant-based diet education	Relapse rates extending to follow-up at 5 years ≤19%, which the authors purport is better than those previously reported. Diet adherence decreased over time.

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
