# Peer review of "Dietary Factors in Sulfur Metabolism and Pathogenesis of Ulcerative Colitis"

_nutrients, 2019, doi:10.3390/nu11040931_

Round 1

Reviewer 1 Report

This is an informative, well organised and well written review. It considers, and amalgamates, diverse concepts in sulphur intake and metabolism and applies them to the clinical problem manifest in the purported association between UC and luminal H2S concentration. I would be very interested in the authors considering the addition of a summary figure that incorporates the broad themes embodied in the heading structure of the manuscript. One imagines a cartoon image of the colon with each of the important sources and mechanisms of production of H2S as well as the H2S clearance mechanisms represented along the course of the organ. The figures included already are descriptive and helpful but having an overview of the key concepts would be very helpful for the reader.

Author Response

The following is a point-by-point responses to each of Referee 1’s comments.

Reviewer 1: This is an informative, well organised and well written review. It considers, and amalgamates, diverse concepts in sulphur intake and metabolism and applies them to the clinical problem manifest in the purported association between UC and luminal H2S concentration. I would be very interested in the authors considering the addition of a summary figure that incorporates the broad themes embodied in the heading structure of the manuscript. One imagines a cartoon image of the colon with each of the important sources and mechanisms of production of H2S as well as the H2S clearance mechanisms represented along the course of the organ. The figures included already are descriptive and helpful but having an overview of the key concepts would be very helpful for the reader.

Response to Reviewer:

We appreciate the reviews comments.  While such a summary figure would be useful, we do have most of the information noted already in the current figures, such as the important sources of H2S (Figure 2) and the trend of detoxification across the colon (Figure 3).  Each of these figures were created de novo and unfortunately it would not be possible to create such a detailed summary figure in the allotted timeframe for review.  We hope the reviewer can appreciate the work in creating the current figures.  

Reviewer 2 Report

I enjoyed reading this hypothesis paper that seeks to link sulfur metabolosm with the pathogenesis of UC. The authors are quite persuasive but I have a few points that I would liek them to address:

Appendicectomy, for appendicitis, is a clearly protective against UC. How might this fit into the hypothesis? 

Cessation of tobacco smoking appears to initiate UC: how might this fit into the hypothesis?  

Is there truly a gradiation in the metabolism of sufur? Is this dictated by the mucosa or bacteria? How might this explain the i) the relatitively fixed disease extent in individuals? ii) How might to expalin left sided disease in some patients and extensive in others? More detail is needed, the section on page 8 is too limited.

What is happening in as countries become Weternised? Why do they get UC? Can you same more about the impact of diet and the enviroment on the changing epidemiology. Please refer to the huge increase in India and South Korea.

I would be fascianted to read a revision.

Author Response

Responses to Referee 2:

We appreciate Referee 2’s careful review of our paper and feel that suggested revisions have greatly improved the manuscript. Below we offer a point-by-point response to each of Referee 2’s comments.

Reviewer 2: I enjoyed reading this hypothesis paper that seeks to link sulfur metabolism with the pathogenesis of UC. The authors are quite persuasive but I have a few points that I would like them to address:

Response to Reviewer: We appreciate the referee’s comments on the merits of the manuscript.

Reviewer 2: Appendicectomy, for appendicitis, is a clearly protective against UC. How might this fit into the hypothesis? 

Response to Reviewer: We agree with the reviewer and feel that its role would be through the gut microbiota. We specifically singled out Fusobacterium spp as playing a part in this and added the following on Page 7, Lines 260-267:

Prevention of excessive cysteine degradation to H2S may underlie the possible protective role of appendectomy for appendicitis in UC [95]. Although the mechanism underlying the purported beneficial effect of appendectomy in UC has yet to be elucidated, one possibility is related to the potential role of the appendix as a reservoir for gut microbes [95]. Specifically, the appendix contains Fusobacterium spp in the healthly state [96] and appendicitis is associated with an abundance of this bacteria [97]. Fusobacterium spp contain a number of proteins necessary to metabolize L-cysteine to H2S [98,99] . Therefore, removal of the appendix, and subsequently a source of H2S production, may contribute to its possible therapeutic role in prevention of UC.

Reviewer 2: Cessation of tobacco smoking appears to initiate UC: how might this fit into the hypothesis?  

Response to Reviewer: There is a well-articulated hypothesis that we have added a reference to in the “H2S Clearance” section of the manuscript. Page 8, Lines 306-307: “Interestingly, this pathway may be facilitated by cyanide, which has been proposed by Levitt and colleagues to explain the well-documented beneficial role of smoking in UC [113]”

Reviewer 2: Is there truly a gradiation in the metabolism of sulfur? Is this dictated by the mucosa or bacteria? How might this explain the i) the relatively fixed disease extent in individuals? ii) How might to explain left sided disease in some patients and extensive in others? More detail is needed, the section on page 8 is too limited.

Response to Reviewer: We thank the reviewer for the comment and we have addressed this in section on page 9, line 337-339 as follows: “The H2S detoxification capacity and expression of enzymes involved in the H2S detoxification pathway (SQR, ETHE1, and TST) have been shown to follow a general trend of highest in the proximal colon to lowest in the rectum [122,123]” . This references a documented enzyme expression pattern as it relates to mucosal detoxification of hydrogen sulfide. It is a single variable of a multivariate hypothesis (Figure 3). It would be expected to particularly contribute to left-sided disease based on expression pattern, but may contribute to more extensive disease if H2S overwhelms detoxification capacity. No changes were made to the manuscript at this time.

Reviewer 2: What is happening in as countries become Westernized? Why do they get UC? Can you same more about the impact of diet and the environment on the changing epidemiology. Please refer to the huge increase in India and South Korea.

Response to Reviewer: We agree with the importance of specifically stating what changes associated with westernization contribute to UC. We have added a comment in the introduction (Page 1, Lines 41-44) that was specifically intended to address the specific diet changes expected with westernization, “Conceptually, the epidemiological data that demonstrates an increasing prevalence of IBD with westernization lends support to this notion as westernization generally results in a transition from a plant-based to animal-based diet [9].”
